# SPROUTY2, a Negative Feedback Regulator of Receptor Tyrosine Kinase Signaling, Associated with Neurodevelopmental Disorders: Current Knowledge and Future Perspectives

**DOI:** 10.3390/ijms252011043

**Published:** 2024-10-14

**Authors:** Nidhi Puranik, HoJeong Jung, Minseok Song

**Affiliations:** Department of Life Sciences, Yeungnam University, Gyeongsan 38541, Republic of Korea; nidhipuranik30@gmail.com (N.P.); jhj010830@naver.com (H.J.)

**Keywords:** ERK/MAPK pathway, GRB2, receptor tyrosine kinase, feedback inhibition, neurodevelopmental disorder, growth factor

## Abstract

Growth-factor-induced cell signaling plays a crucial role in development; however, negative regulation of this signaling pathway is important for sustaining homeostasis and preventing diseases. SPROUTY2 (SPRY2) is a potent negative regulator of receptor tyrosine kinase (RTK) signaling that binds to GRB2 during RTK activation and inhibits the GRB2-SOS complex, which inhibits RAS activation and attenuates the downstream RAS/ERK signaling cascade. SPRY was formerly discovered in *Drosophila* but was later discovered in higher eukaryotes and was found to be connected to many developmental abnormalities. In several experimental scenarios, increased SPRY2 protein levels have been observed to be involved in both peripheral and central nervous system neuronal regeneration and degeneration. SPRY2 is a desirable pharmaceutical target for improving intracellular signaling activity, particularly in the RAS/ERK pathway, in targeted cells because of its increased expression under pathological conditions. However, the role of SPRY2 in brain-derived neurotrophic factor (BDNF) signaling, a major signaling pathway involved in nervous system development, has not been well studied yet. Recent research using a variety of small-animal models suggests that SPRY2 has substantial therapeutic promise for treating a range of neurological conditions. This is explained by its function as an intracellular ERK signaling pathway inhibitor, which is connected to a variety of neuronal activities. By modifying this route, SPRY2 may open the door to novel therapeutic approaches for these difficult-to-treat illnesses. This review integrates an in-depth analysis of the structure of SPRY2, the role of its major interactive partners in RTK signaling cascades, and their possible mechanisms of action. Furthermore, this review highlights the possible role of SPRY2 in neurodevelopmental disorders, as well as its future therapeutic implications.

## 1. Introduction

Many cellular decisions and actions essential for the proper and strong growth of multicellular animals are mediated via cell-to-cell communications. Growth factors (GFs), which attach to and activate cell-surface receptors to relay cues intracellularly, contain several signals encoded in them. Receptor tyrosine kinases (RTKs) are one of the main cell-surface receptor superfamilies [1]. The RTK protein superfamily is well known for its ability to promote cell proliferation; however, it also plays important roles in eukaryotic development and homeostasis. The maintenance and survival of both developing and adult tissues are among these functions, as are the control of cell shape changes during migration and morphogenesis. These include the patterning of cells and tissues. Tyrosine residues are selectively trans-autophosphorylated, and RTK activity increases when GF binds to the receptor in the extracellular area. Although some of these sites serve as docking sites for other adaptor/effector scaffold proteins and enzymes, others are crucial for maintaining the kinase in its active state. Natural ligands of the RTKs are used to determine their names. Membrane proteins, such as the epidermal growth factor receptor (EGFR) family [2], fibroblast growth factor receptor (FGFR), platelet-derived growth factor receptor (PDGFR) family, vascular endothelial growth factor receptor (VEGFR) family, Trk subfamily (nerve growth factor [NGF], brain-derived neurotrophic factor [BDNF] [3], and neurotrophin [NT]), and insulin receptors (insulin, insulin-like growth factor [IGF]) [4], have been categorized according to their structural and ligand affinity features [4,5,6,7]. The MAPK cascades, PLCγ, PI3K, and JAK/STAT signaling pathways are the cascades that are most frequently used to convey the signal from RTKs. JNKs, P38-MAPKs, and RAS/ERK1/2 are examples of MAPK cascades. In general, JNKs and p38 MAPKs react to inflammatory cytokines and environmental stress, whereas ERKs regulate cell proliferation, growth, and differentiation through their transcriptional targets [8,9,10,11]. These signaling cascades are strictly regulated by distinct mechanisms, as hyperactivation of these pathways causes diseases such as cancer, developmental problems in multiple systems, including the nervous system, and learning difficulties. Negative feedback regulation of the cascade is one of the most prevalent processes [12,13]. A basic representation of FGF signaling-induced gene expression for cell proliferation, differentiation, and feedback inhibitors is shown in Figure 1. Figure 1 depicts that FGFs interact with their specific receptors, facilitating receptor dimerization and autophosphorylation. This activation process recruits secondary signaling molecules, including PLCγ, FRS2α, and GRB2, which contribute to the formation of signaling complexes. The interaction of FGFs with their receptors activates three primary downstream signaling pathways: the Ras/MEK/MAPK/ERK pathway, the PI3K/AKT pathway, and the PLCγ pathway. The MAPK/ERK pathway subsequently translocates to the nucleus, where it regulates gene expression related to cell proliferation, differentiation, and feedback inhibition.

The SPROUTY (SPRY) family and SPRED proteins are important negative regulators of RAS signaling. The C-terminal SPR domains of SPRED and SPRY are conserved, although their mechanisms for controlling RAS signaling are different. According to a previous study, in a mouse model of severe truncation of the forebrain, the head tissues derived from the cephalic neural crest and the lungs developed when both SPRY2 and SPRY4 functions were abated. This discovery provided clear genetic evidence that SPRY2 and SPRY4 are crucial for vertebrate head and lung development. Among all RTK signaling feedback inhibitors, this review will discuss SPRY2 in detail.

In *Drosophila melanogaster*, the SPRY protein was primarily recognized as an antagonist of ERK1/2 signaling and inhibitor of RAS signaling downstream of various RTKs during different morphological development processes, as well in the oogenesis process [15]. SPRY2 binding to the adapter molecule GRB2 leads to phosphorylation; however, the particular regulatory kinase(s) involved are not yet known. In *Drosophila*, dSPRY have a different inhibitory target such as inhibiting the activation of MAPK upstream of RAS in eye development and inhibiting MAPK downstream of RAS during wing development [16]. Further studies revealed that vertebrates also have SPRY genes, and vertebrate SPRY members can likewise inhibit RTK signaling, the same as dSPRY. Hanafusa et al. (2002) [17] studied the RAS/MAPK signaling pathway via the inhibitory mechanism of SPRY2 in mice and Xenopus and showed that overexpression of mSPRY2 represses FGF-mediated limb growth in chickens. The results of their experiments indicated that tyrosine (Tyr) phosphorylation, which can be induced by RTK signaling, activates the inhibitory functions of SPRY1 and SPRY2 [17]. Thus, SPRY2 is involved in a typical negative feedback loop and is now recognized as a conserved inhibitor of RTK signaling in higher eukaryotes. However, the precise chemical mechanism underlying SPRY2 activity remains unclear.

SPRY2 also plays a crucial role in human embryonic stem cell (hESC) self-renewal, with low levels reducing survival and proliferation. However, SPRY2 knockdown cells remain responsive to growth factors, showing increased cell numbers in response to higher concentrations of FGF2 and EGF [18].

Further studies revealed that humans have four SPRY genes that code for four different isoforms of the SPRY protein, SPRY1, SPRY2, SPRY3, and SPRY4, which are homologs of dSPRY; however, *Drosophila* has only one SPRY protein. Human SPROUTY (hSPRY) isoforms have variable N-terminal sequences but share considerable cysteine (Cys)-rich sequence homology at their C-termini, as shown in Figure 2.

Members of the human SPRY family exhibit highly restricted expression patterns during the early stages of embryonic development. Furthermore, their expression is closely correlated with known FGF signaling sites, suggesting that they may also play a role as negative regulators of FGF signaling during vertebrate development. To further explore the SPRY family, we have used various online available tools as mentioned in [19]. The UniProt database (https://www.uniprot.org/) served as the primary resource for retrieving detailed information on the functional regions and domain proteins. The UniProt database (https://www.uniprot.org/) was used to retrieve the SPRY protein-related information (accessed on 3 October 2024) and sequence. In addition, the UniProt alignment tool was utilized to determine the percentage identity of SPRY2 compared to each SPRY protein 1, 3, and 4 and phylogenetic distance. This approach facilitated a comprehensive analysis of sequence similarity and functional conservation among the SPRY family shown in Figure 3. The SPRY protein family shows around 40–55% on the similarity index. The SPRY2 protein exhibits low tissue specificity, being expressed in nearly all major tissues. Analysis of SPRY2 expression across various tissues was conducted using the Genotype-Tissue Expression (GTEx) database (https://www.gtexportal.org/). The results indicate that while SPRY2 is present in almost all tissues, it is expressed at notably higher levels in the cerebellum and cerebellar hemispheres of the brain (Figure 4).

A basic comparison of all four human SPRY proteins and their roles in the RTK signaling pathway is presented in Table 1.

SPRY proteins control RTK signaling according to the circumstances. In particular, SPRY proteins function as negative feedback regulators of ERK signaling by preventing different RTKs from activating RAS upstream. A conserved Tyr residue is phosphorylated, and the SH2 domain of GRB2 docks at the N-terminus of SPRYs. This mechanism of control implies that SPRYs tether GRB2 away from the SOS, thereby impairing RAS activation. Interaction between SPRY proteins and CRAF has been demonstrated; however, the ramifications of this interaction are still being studied. By upregulating SPRY proteins and dual-specificity phosphatases, activation of the RAS/MAPK pathway under normal physiological conditions triggers ERK1/2-mediated negative feedback signaling, which suppresses the activation of RTKs and RAS [20,21].

SPRY proteins block signaling from different GF receptors, such as glial cell-derived neurotrophic factor (GDNF), VEGF, FGF, NGF, and PDGF. The ability of SPRY proteins to block signaling pathways is crucial for regulating cell survival and proliferation. Numerous methods for these reactions have been identified, including direct interactions with molecules known to be regulators or effectors of this signaling cascade, such as RAF1, GRB2, SHP2, and others. Recent research has also shown that SPRY proteins influence other biological pathways and signals, such as PI-PLC, which explains their regulatory effects on T-cell proliferation and calcium-mediated signaling. Notably, no studies on the effects of SPRY proteins on other MAP kinase cascades are available, and it appears that their effects on the GF-activated MAP kinase pathways are restricted by the MEK/ERK pathway [22].

### 1.1. hSPRY2 Protein Structure and Their Interacting Partners

To understand the functions of SPRY2 and their interactions with other proteins, the protein–protein interactions of closely related proteins were projected using the online String tool (https://string-db.org/). However, we focused exclusively on proteins for which published data demonstrate a role in neurodevelopment. This approach ensures that our discussion is grounded in established research.

All hSPRY proteins have a highly conserved C-terminal Cys-rich region and a variable N-terminal region, and both terminals have various phosphorylation sites. The structure of hSPRY2 and the interaction sites for various molecules are shown in Figure 5. The biological activity of SPRY is regulated by phosphatases such as PTEN, PP2A, Src homology-2-containing SHP2, and PTP1B, and kinases such as DYRK1A, TESK1, and Mnk1. To regulate ubiquitination and SPRY2 degradation, the ubiquitin ligases c-Cbl and SIAH2 interact with the N-terminus. Overexpression of SPRY2 promotes ERK signaling in non-neuronal cells expressing EGF receptors by binding to and sequestering c-Cbl, which prevents EGFR ubiquitylation and destruction. Furthermore, SPRY2 may use its scaffolding ability to serve as an adapter protein for c-Cbl or to target Cbl for ubiquitination by other proteins but not via its E3 ligase function [23]. The interacting proteins and their roles are summarized in Table 2.

The analysis conducted using the IntAct database (https://www.ebi.ac.uk/intact/) revealed a strong physical interaction of SPRY2 with the Cbl and GRB2 proteins, which are associated with RTK signaling (Figure 6).

### 1.2. Molecular Mechanisms of SPRY2

SPRY2 protein is expressed in the brain and various neuronal cell types, including neurons, astrocytes, oligodendrocytes, and neural stem cells. In these cells, SPRY2 plays a key role in modulating signaling pathways that are critical for neuronal development, survival, and function.

GF stimulation causes transcriptional upregulation of SPRY2, which is typically regarded as a negative feedback inhibitor because it blocks the associated pathways. After RTK activation, they move from the cytosol to the cell membrane, where they are phosphorylated at Tyr55 residues and attached via palmitoylation. A simplified schematic illustrates the feedback regulation of FGF/EFGF and PI3K/PIP2 signaling by SPRY2.

For characterization of membrane association of SPRYs, Impagnatiello (2001) [44] performed biochemical and immunofluorescence assays that showed that caveolin-1 associates with both endogenous and overexpressed SPRY-1 and -2 in the perinuclear and vesicular structures, and both SPRYs are tethered to membranes by palmitoylation. They are phosphorylated on Ser residues, and a subset is drawn to the leading edge of the plasma membrane in response to GF stimulation, confirming that mammalian SPRY-1 and -2 are membrane-anchored proteins [44].

Functional activation of SPRY2 requires the phosphorylation of the critical Tyr55 residue by c-Src kinase, as shown in Figure 5a. Mason and the team cotransfected c-Src with Tyr kinase, a prototype member of this protein family, to validate the function of the Src kinase. This resulted in significant Tyr phosphorylation of SPRY2 in serum-starved NIH3T3 cells. By contrast, constitutive phosphorylation of SPRY2 was not induced by cotransfection with catalytically inactive Src, which is unable to bind ATP or phosphorylate the substrates. This study demonstrates that Tyr phosphorylation of SPRY2 requires a Src-like kinase.

SPRY2 is significantly more inhibitory than SPRY1 and 4 and binds to GRB2 through a C-terminal proline-rich region, which is a unique feature of SPRY2. By binding to the adaptor protein GRB2, SPRY2 stops RAS-induced ERK activation upstream. Furthermore, SPRY2/4 interacts with RAF after RAS, and, as a result, SPRY2 obstructed the ERK pathway both before and after RAS.

SPRY2 had a greater inhibitory effect on ERK activation than the other isoforms; however, SPRY2 shows no inhibitory effect when RAS was constitutively activated. In addition to the ERK pathway, SPRY1 and SPRY2 inhibit activation of PLCγ. AKT signaling is also affected by SPRY2 by enhancing the activity of PTEN; however, it is dependent on the cell type and GFs involved in signaling [36]. In adult sensory neurons, downregulation of SPRY2 led to the activation of RAS and pERK in response to FGF-2, whereas phosphorylation of pAKT and p38 remained unchanged [45]. The cytoplasm of SPRY2-deficient peripheral neurons contained activated ERK, but not the nucleus. This could be due to the distinct ERK1/2 inactivating of MAP kinase phosphatase activities, although this is currently conjectural. MKP3 is an enzyme found in the cytoplasm, while MKP1 is found in the nucleus. Consequently, selectively targeting MKP3 could have comparable effects to interfering with SPRY2 in increasing cytoplasmic ERK activation [23].

Tyr55 phosphorylation is required for SPRY2 molecular function, and SPRY2 is translocated to the cytoplasm of the membrane. Despite the lack of clarity regarding the mechanisms regulating this translocation, three scenarios have been proposed. First, the translocation of SPRY2 to membrane ruffles may be caused by the binding of SPRY2 to PIP2, as the C-terminus of SPRY2 has a membrane translocation domain that binds to PIP2. Second, endothelial cells palmitoylate SPRY2 at the C terminus of the protein, which may help SPRY2 to attach to the membrane. However, it is unknown whether palmitoylation of SPRY2 is a dynamic process controlled by GF stimulation in cells. Lastly, following Ser phosphorylation in response to GF stimulation, SPRY2 proteins may translocate to the plasma membranes through interactions with caveolin-1 through their C-terminal domains. However, the inhibition of GF-stimulated cell migration, proliferation, and differentiation by SPRY proteins requires the translocation of SPRY2 [36].

To confirm the translocation of SPRY2, Hanafusa et al. (2002) [17] constructed several deletion mutants of mSPRY2 and analyzed their activities. Coimmunoprecipitation experiments demonstrated the significance of mSPRY2’s conserved C-terminal region for the production of mSPRY2 homodimers. They observed the capacity of the mutant mSPRY2 to translocate to the area of the plasma membrane where wild-type SPRY2 translocates in response to EGF. All other C-terminally deleted mutants of mSPRY2ΔC2 to mSPRY2ΔC6 did not translocate to the ruffling membrane region in response to EGF, but mSPRY2ΔC1 did, albeit less than full-length mSPRY2. They examined the capacity of the deletion mutants to phosphorylate Tyr and found that they exhibited an inhibitory action. In response to EGF, mSPRY2ΔC1 experienced Tyr phosphorylation, while the other mutants did not. Similar outcomes were observed after FGF stimulation. Accordingly, only mSPRY2ΔC1 was able to significantly block FGF-induced Elk-1 activation in the reporter assay. Therefore, a strong link was noted between SPRY2’s capacity to undergo Tyr phosphorylation and function as an inhibitor and its ability to translocate to the plasma membrane. Interestingly, mSPRY2ΔC5-CAAX, the construct that was created when the Cys-Ala-Ala-X motif of RAS was fused to it to localize this mutant to the plasma membrane, underwent Tyr phosphorylation in response to stimulation and prevented Elk-1 from being activated. These findings suggest that Tyr phosphorylation of SPRY2 in response to stimuli may require translocation of the protein to the plasma membrane, which is necessary for its inhibitory function.

According to Yim et al. (2015) [16], ectopic overexpression of SPRY2 reduces VEGF- and FGF-induced ERK activation but not EGF-induced activation. After FGF stimulation, the SPRY2 Tyr55F mutant is unable to suppress ERK activation. The N-terminal SH3 domain of GRB2 (residues 59–64) and two proline-rich stretches of SPRY2 (303–307) interact with GRB2 constitutively; however, the inhibitory effect of SPRY2 on FGF-induced ERK activation is independent of its ability to bind GRB2 and is unaffected by RAS-GTP loading. On the other hand, SPRY2 lessens RAF activation. SPRY2 is mostly found in caveosomes and vesicular/endosomal structures but also in the plasma membrane. SPRY2 obstructs the process of early-to-late endosomes and influences the trafficking of activated EGFR [46].

Ozaki et al. (2001) [47] studied the correlation between ERK activation and SPRY gene expression in Swiss 3T3 mouse fibroblasts stimulated with GFs and mitogenic agents and in human tumor cells showing the constitutively activated ERK pathway. All reagents that activate ERKs induced SPRY gene expression and activated Raf-1 kinase. Furthermore, pretreated mouse fibroblasts and human tumor cells with MEK inhibitors suppressed SPRY expression. These findings indicate that SPRY gene expression is positively regulated by the ERK.

Activated PP2A partially resists the resulting increase in phosphorylation by attaching to the N-terminal Cbl-TKB-binding motif. Notably, two other ubiquitin E3 ligases attach to the N-terminus of SPRY; SIAH2 binds constitutively to a distinct location and c-Cbl binds strongly to the TKB-binding motif. Both proteins can control the ubiquitination and degradation of SPRY proteins. Evidence suggests that SPRY proteins are significantly altered covalently to regulate their position, stability, affiliation, and elimination [24].

DaSilva et al. (2006) [33] examined the stability of SPRY2 through ubiquitination and proteasomal degradation and found that Mnk1 kinase phosphorylates Ser112 and Ser121 on SPRY2, which in turn controls the phosphorylation of Tyr55. Moreover, inhibition of Mnk1 activity and substitution of Ser112 and Ser121 by Ala increases the degradation of SPRY2 confirm the role of critical Ser phosphorylation in SPRY2 stability. Lao et al. (2007) [26] demonstrated that several Ser or Thr residues in a Ser/Thr-rich region on N-terminal are responsible for the slower migration of SPRY2 bands in response to GF stimulation. However, no evidence indicates that casein kinase 2 phosphorylates SPRY, even though SPRY have a consensus site for this enzyme, while PP2A dephosphorylates at least two of the phosphorylated Ser residues.

DYRK1A phosphorylates and interacts with SPRY2 at Thr75. The relationship between DYRK1A and SPRY2 phosphorylation at Thr75 appears to have an adverse effect on SPRY2’s ability to operate as an antagonist of RTK signaling, as evidenced by the improved ability of SPRY2 to inhibit Erk1/2 activation by FGF when Thr75 was substituted with Ala. However, further research is needed to confirm the significance of Thr75 phosphorylation, such as DYRK1A silencing or deletion experiments. Similarly, it has been proposed that the FGF-induced phosphorylation of Tyr227, but not EGF-induced phosphorylation, enhances the capacity of SPRY2 to block the FGF-triggered ERK1/2 cascade [28].

Chandramouli et al. (2008) [34] demonstrated that TESK1 interacts with SPRY2 to direct the protein to vesicular compartments such as endosomes. The ability of SPRY2 to block GF activity was attenuated by TESK1 regardless of its kinase activity. This is mainly due to TESK1 interfering with SPRY2/GRB2 connections and PP2A dephosphorylating Ser residues. Similar to c-Cbl, the ring finger domain of SIAH2 binds to the N-terminal region of SPRY2 without Tyr55 phosphorylation and leads to proteasomal degradation of SPRY2 [31]. Another study demonstrated that a dominant-negative SIAH2 ring finger mutant increased the amount of SPRY2 in SW1 melanoma cells and decreased tumorigenesis and metastasis, which is consistent with the role of SIAH2 in controlling the concentration of SPRY proteins [30].

The aforementioned studies demonstrate that SPRY2 not only regulates the RTK signaling pathway but also modulates additional critical pathways involved in cell growth and development. The FGF/EFGF and PI3K/PIP2 signaling pathways are shown in Figure 7, while a comprehensive overview of the role of SPRY2 across various signaling pathways, along with its mechanisms of action, is summarized in Table 3.

### 1.3. SPRED2, a SPROUTY-Like Protein

SPRED is a SPRY-like protein with an N-terminal EVH-1 domain, a central c-Kit-binding domain, and a C-terminal SPR domain. They have a negative regulatory effect on RTK signaling by inhibiting the RAS–MAPK pathway. SPRED1 directly binds to neurofibromin, RAS-GAP, and c-KIT. Neurofibromin function depends entirely on this interaction. Mutations causing loss-of-function in SPRED1 have been found in human tumors, resulting in Legius syndrome, a developmental disease. Mice with genetically altered SPRED genes exhibit behavioral abnormalities, dwarfism, and several other characteristics, such as an elevated risk of leukemia [57]. Members of the SPRED family have a C-terminal SPR highly similar to the C-terminal region of SPRY proteins. SPRED proteins are activated by GF-mediated MAP kinase activation and cytokine receptor-induced activation of ERK [46]. One study examined the ability of SPRY and SPRED to prevent ERK activation caused by stimuli other than RTK agonists, such as prostanoids and cyclopentenone. These findings revealed both similar and unique characteristics of how GFs and cyclopentenone prostanoids activate RAS-dependent pathways. Furthermore, they offer the first evidence that SPRY and SPRED are negative regulators of the activation of the ERK/Elk-1 pathway caused by reactive lipid mediators, as well as GFs [46].

The functional domains and amino acid sequences of SPRED2 and SPRY2 and a general schematic representation of feedback inhibition of the RAS/MAPK/ERK pathway are shown in Figure 8. Myocyte and neuronal cell development depend on the RAS/MAP kinase pathway. Endogenous SPRED controls differentiation in these cell types, as demonstrated by the expression of a dominant-negative version of SPRED or by injecting a SPRED antibody. Although SPRED is constitutively linked to the RAS, it does not inhibit RAS activation or RAF membrane translocation. By contrast, SPRED suppressed the phosphorylation and activation of RAF, which in turn decreased MAP kinase activation. Possibly, it could be said that SPRED may be a family of proteins that regulates MAP kinase signaling and RAS ± RAF interaction [58].

### 1.4. SPRY2 in Neurodevelopment

Cell signaling of GFs plays a crucial role in the development of flexibility and is essential for adult plasticity, including memory development. GFs mediate overlapping functional endpoints by converging intracellular cascades (e.g., RAS/MEK/MAPK), even though they interact with receptors with different kinase domains. The central and peripheral nervous systems express neurotrophins and other GFs in limited quantities, which regulate the number and types of neurons required for the proper density of dendritic fields [59]. Additionally, these GFs target innervation and support cell survival, neurogenesis, differentiation, axonal outgrowth, synaptogenesis, and activity-dependent synaptic pruning [60]. Among other RTKs, Trk receptor-BDNF mediated and FGF-mediated signaling have been the most studied in nervous system development and neurodevelopmental disorders.

FGFs are essential signaling molecules involved in the growth, maintenance, and repair of the brain. According to Manson I, 2007 [61], and Lahti et al. (2011) [62], FGF affects the complex interactions between myelinating cells and axons, as well as the connections between astrocytic and microglial processes and axon guidance and synaptogenesis. By activating FGFR1, FGF2 may play a significant role in neuropsychiatric disorders. Its suggested uses include anxiolytic, antidepressant, and memory-enhancing properties [63]. Repeated electroconvulsive therapy in humans is a highly effective treatment for depressive disorders because it stimulates glial cell proliferation and neurogenesis in the hippocampal region. However, SPRY2 downregulates neurogenesis by blocking RTK signaling. The findings of this study indicate that ECS modifies SPRY2 expression in prefrontal cortex cells that are SPRY2 immunoreactive, while also reducing SPRY2 expression in glial cells and promoting glial cell proliferation [64]. This finding was corroborated a recent study by Dow et al. (2015) [65], who demonstrated that antidepressant-like effects could be produced by disrupting SPRY2 activity in the dorsal hippocampus. This study focused on the relationship between neuroplastic alterations in the brain and diseases related to stress and depression. Under these circumstances, FGF2 signaling causes neurogenesis, which is inhibited in the hippocampal region. SPRY2 blocks FGF signaling. They used a viral vector-mediated method to disrupt SPRY2 activity and investigate the impact of SPRY2 on neurogenesis in rats. They discovered that this disruption of SPRY2 function in the dorsal hippocampus could affect molecular behavioral metrics, similar to those of antidepressants.

While many signaling pathways are known to be involved in PNS regeneration, the PI3K/Akt and ERK/MAPK pathways are thought to be particularly significant because of their critical functions in controlling cell survival, growth, division, and proliferation. For instance, neurotrophic factor-mediated axonal regeneration and neuronal survival depend on the PI3K/Akt and/or ERK/MAPK pathways. ERK signaling plays a crucial role in the axonal expansion of adult mouse dorsal root ganglia (DRG) neurons activated by NGF or GDNF, as well as in mediating the protective effects of BDNF to support neuronal survival following nerve injury. Neurotrophic factor-independent survival of adult DRG and sympathetic superior cervical ganglia neurons is preserved by PI3K signaling. Pharmacological inhibition of PI3K decreases GF-dependent neurite extension in dissociated adult rat DRG neurons, suggesting that PI3K promotes axonal outgrowth [66].

Taketomi et al. (2005) [67] examined the condition of the enteric neural system in SPRY2^-/-^ mice, and it was shown that esophageal achalasia and intestinal pseudo-obstruction were caused by enteric nerve hyperplasia in mice lacking the SPRY2 gene. GDNF induces ERK and Akt hyperactivation in enteric nerve cells. When SPRY2-deficient animals were administered with anti-GDNF antibodies, nerve hyperplasia was reversed. These findings indicate that SPRY2 is a negative regulator of GDNF in the survival or development of enteric nerve cells in neonates. Subsequent investigations revealed that the colon ganglia of SPRY2^−/−^ mice had far greater ERK and Akt activation than those of wild-type mice.

Appropriate neural system development depends on ERK/MAPK signaling. Through the ERK/MAPK signaling pathway, a number of signaling molecules that control the well-coordinated processes of neurodevelopment transduce developmental information. The ERK/MAPK pathway is a possible new therapeutic target for a number of neurodevelopmental disorders. Nevertheless, the precise mechanism by which the ERK/MAPK signaling pathway elicits certain responses in neurodevelopment remains largely unknown after years of research [68]. The function of SPRY2, which is upregulated in various experimental models of neuronal degeneration and regeneration, was succinctly described by Hausott and Klimaschewski in 2019 [23]. Owing to its increased production in pathogenic settings, SPRY2 is a desirable pharmaceutical target for improving intracellular signaling activity in stimulated astrocytes or damaged neurons, particularly in the ERK pathway. Further studies on neural lesion models, such as kainic acid-induced epilepsy or endothelin-induced ischemia, have verified that the reduction in SPRY2 proteins promotes the proliferation of activated astrocytes and thus lowers secondary brain damage. Moreover, in peripheral nervous system lesions, SPRY2 downregulation enhances nerve regeneration. When combined, targeting SPROUTYs as intracellular ERK pathway inhibitors has enormous potential for the treatment of various neurological conditions, including gliomas. The absence of enzymatic activity in proteins makes it challenging to create chemical compounds that can precisely and directly modulate SPRY function. On the other hand, blocking SPRY expression through gene therapy or small interfering ribonucleic acid (siRNA) treatment offers a practical way to assess the therapeutic potential of promoting ERK activity indirectly in neurological disorders.

Downregulation of SPRY2 promotes the formation of elongated axons in cultured peripheral and central neurons. The study conducted by Marvaldi et al. (2015) [69] examined the effects of SPRY2 global knockout mice on peripheral axon regeneration in vivo and axon outgrowth in vitro. Dissociated neurons from adult sensory ganglia lacking SPRY2 exhibited increased axon outgrowth and greater activation of extracellular signal-regulated kinases. Heterozygous SPRY21/2 neuronal cultures show prominent axonal extension, whereas homozygous SPRY22/2 neurons mostly display a branching phenotype. In motor, but not sensory, testing paradigms, SPRY21/2 mice recovered more quickly after sciatic nerve crushing. The higher densities of motor endplates in the muscles of the hind limbs, greater number of myelinated fibers in the regenerated sciatic nerve, and elevated amounts of GAP-43 messenger ribonucleic acid (mRNA), a downstream target of extracellular regulated kinase signaling, were the reasons given by the study for the improvement in the rotarod test. By contrast, homozygous SPRY22/2 animals exhibit improved mechanosensory function, increased epidermal innervation, a higher number of non-myelinated axons, and a greater number of IB4-positive dorsal root ganglia neurons. The current findings suggest that SPRY2 is a novel potential target for pharmacological inhibition to speed up long-distance axon regeneration in injured peripheral nerves and validate the functional significance of receptor Tyr kinase signaling inhibitors for axon outgrowth during development and nerve regeneration [69].

Neurotrophic factor-mediated axon regeneration requires the RAS/RAF/ERK and PI3K/Akt pathways. While active Akt enhances axon diameter and branching, activation of RAF is required for axon growth by DRG neurons throughout development and causes axonal elongation in sensory neurons [70]. ERK signaling is involved in gene regulation; however, it is also necessary for regenerative signaling in DRG neurons to be transported retrogradely and for local axon building brought on by neurotrophins [71]. Axon growth is also linked to SPRY2, and new experimental findings have shown that endogenous RTK signaling inhibitors, such as SPRY2, prevent axon regeneration in DRG neurons. Hausott et al. (2009) observed that the knockdown of SPRY1, -2, and -4, by siRNA technology in cell cultures positively regulated axon growth, followed by increased activation of ERK and RAS caused by FGF-2, but overexpression of SPRY2 prevented axon development. These investigations have verified SPRY’s function in the growth and development of neurons [45].

According to current hypotheses on the pathophysiology of schizophrenia, individuals with this disorder may have altered brain plasticity, including diminished neuronal migration and proliferation, delayed myelination, and incorrect synaptic modeling. Although many anomalies associated with schizophrenia are linked to functional changes in BDNF that are critical for neuroplasticity, the regulatory mechanisms or mechanisms causing aberrant BDNF signaling in schizophrenia remain unclear. In 2008, Pillai A [72] examined the possibility that the aberrant expression of the GF signaling regulator SPRY2 was linked to alterations in BDNF mRNA levels in schizophrenia. Adult rats were used to assess the effects of antipsychotic medications on SPRY2 expression. Dorsolateral prefrontal brain samples from the Stanley Array Collection were used to examine the expression of SPRY2 and BDNF genes. In both bipolar disorder and schizophrenia, there is a considerable decrease in the expression of BDNF and SPRY2, according to quantitative real-time polymerase chain reaction analysis of RNA in 100 individuals. Furthermore, a strong association was observed between these two genes in patients with bipolar disorder, schizophrenia, and controls. Rat frontal cortex SPRY2 and BDNF mRNA and protein levels were affected differently by long-term treatment with antipsychotic medications such as olanzapine and haloperidol. The results of this study, which show a correlation between alterations in BDNF and decreased expression of SPRY2, raise the idea that the latter is the result of treatment-related factors rather than important elements in the pathophysiology of bipolar illness or schizophrenia. Additional research on the signal transduction pathways connected to SPRY2 may aid in the development of novel treatment plans for these conditions.

Overall, a variety of in vitro and in vivo studies demonstrate the potential of SPRY2 in regulating normal neuronal cell development. Alterations in the expression or mutations of SPRY2 proteins can disrupt signaling cascades, leading to neurodevelopmental disorders. A comprehensive summary of the key in vitro and in vivo studies is presented in Table 4.

### 1.5. SPROUTY2: Challenges and Unresolved Questions

Combining the available data, SPRY2 appears to be tightly regulated at the expression level and through several covalent changes, which contribute to its significant role in RTK signaling regulation and neurodevelopment. Stimulation of at least one of the circuits on which it feeds back results in this expression. Evidence suggests that, once expressed, SPRY2 is in an inaccessible/inactive confirmation that needs to be correctly phosphorylated and then dephosphorylated at Ser residues for proper activation. SPRY2 is present for a short time, as evidenced by the fact that SPRY proteins are targets of ubiquitination and are eventually eliminated by Cbl and SIAH2. Additionally, there appears to be some competition between the downregulators (c-Cbl) and activators (PP2A) of SPRY2. However, the following question still arises: Are SPRY proteins the major targets of Cbl-directed ubiquitination? If they are, then this suggests that it is desirable to confine SPRY activation to a defined temporal window. According to Rubin et al. (2005) [28], Cbl may have two roles. In addition to ubiquitylating SPRY-associated proteins and sorting them for proteasomal or lysosomal degradation, Cbl may also be involved in the ability of SPRY to disrupt signaling pathways because it is a multivalent adaptor that can engage more than 50 different proteins.

According to the numerous studies discussed above, the active state of the associated kinases allows SPRY proteins to perceive the intracellular environment. This is possible through a variety of kinases and signaling pathways. A significant concern arises if the different kinases and phosphatases mentioned above change the state of preparedness of SPRY2 proteins. Which proteins are SPRY2 bringing into a complex to determine its primary function? Numerous studies lend credence to the theory that GRB2 proteins modulate the processes governing hSPRY2 activation in various tissues or at various developmental stages, rather than recruiting hSPRY2 to the plasma membrane or directly contributing to the negative regulation of the Erk–Elk1 pathway via hSPRY2.

## 2. Conclusions and Future Prospective

The RTK pathway plays a critical role in almost every stage of brain circuit development and is linked to many neurodevelopmental disorders. ERK, a crucial regulator of cell proliferation, exists in all cells and tissues. SPROUTY2 is expressed in brain tissue and serves as a critical regulatory protein of the RTK signaling pathway. Studies have demonstrated that SPRY2 is upregulated in various experimental models of neuronal degeneration and regeneration. Its increased expression under pathological conditions positions SPRY2 as a promising pharmacological target for enhancing intracellular signaling activities, particularly within the ERK pathway, in affected neurons and activated astrocytes.

In conclusion, SPRY2 is an effective regulatory protein of the FGF/VEGF/EGF/BDNF signaling pathway and could be targeted as a therapeutic molecule in neurodevelopmental disorders.

## Figures and Tables

**Figure 1 ijms-25-11043-f001:**
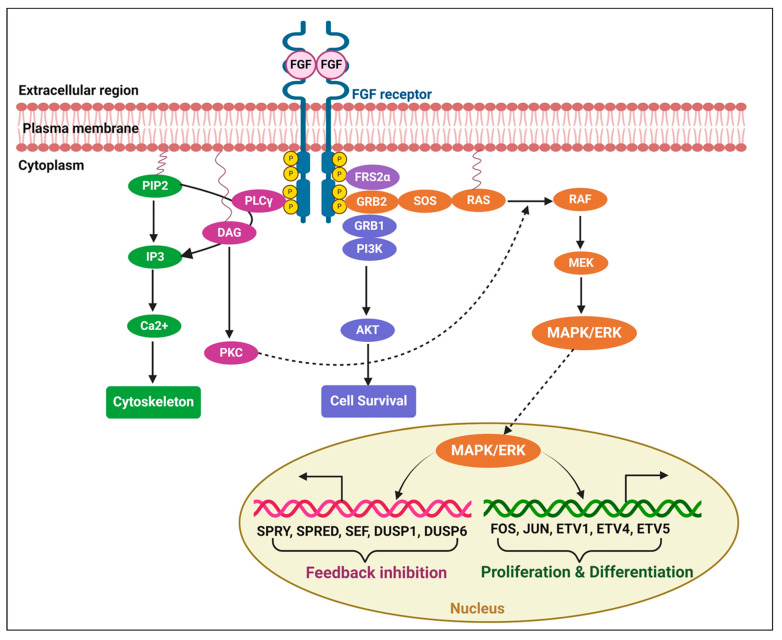
Fibroblast growth factors (FGFs) interact with the FGF receptor domain and facilitate receptor dimerization autophosphorylation, which in turn attracts other secondary signaling molecules such as Phospholipase C Gamma (PLCγ), fibroblast growth factor receptor 2 alpha (FRS2α), and Growth factor receptor-bound protein 2 (GRB2) and assembles them in signaling complexes. A representation of the three primary downstream signaling complexes—the Ras/MEK/MAPK/ERK, PI3K/AKT, and PLCγ pathways—is shown. MAPK/ERK is translocated in the nucleus and activates the gene expressions responsible for cell proliferation and differentiation, as well as for feedback inhibition. (The figure concept was adopted from Diez del Corral and Morales, 2017 [14]), modified and recreated by BioRender.

**Figure 2 ijms-25-11043-f002:**
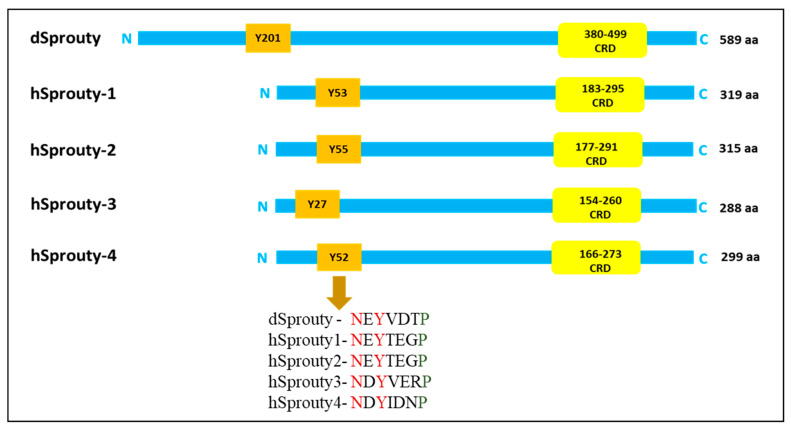
Comparative structure of *Drosophila* SPROUTY and human SPROUTY 1, 2, 3, and 4. All SPROUTY proteins have conserved tyrosine residue on the N-terminal for the activation and functioning of SPROUTY protein. Moreover, dSPROUTY and hSPROUTY also have cysteine-rich domains on the C-terminal that have binding sites for Raf and other proteins, which leads to inhibition of RTK signaling. (Figure was created on PowerPoint).

**Figure 3 ijms-25-11043-f003:**
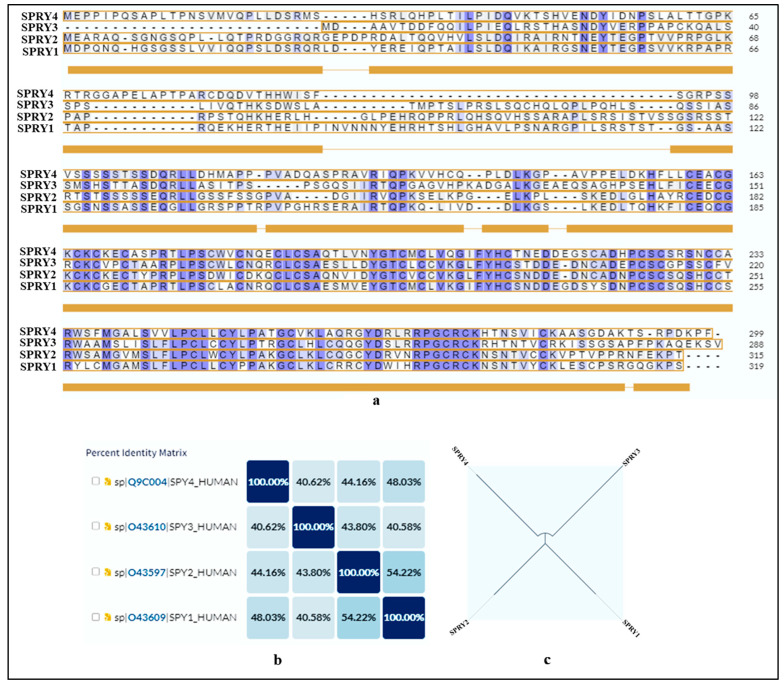
Sequence similarity between SPROUTY proteins. (**a**) Comparison of the alignment of the amino acid sequence of the SPROUTY 1, 2, 3 and 4 proteins (similar amino acid sequences are highlighted); (**b**) percentage identity; and (**c**) phylogenetic tree.

**Figure 4 ijms-25-11043-f004:**
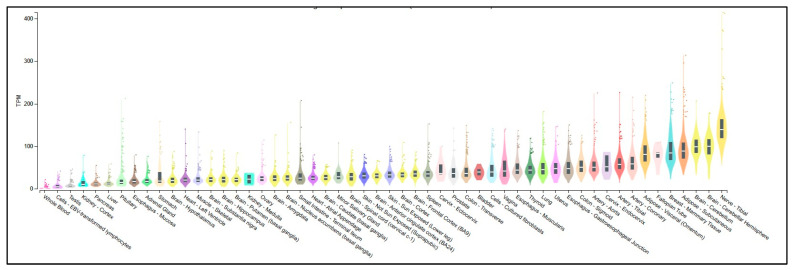
Bulk tissue expression of SPRY2 protein across various tissues. The data indicate the presence of SPRY2 in all examined tissues, with notably higher expression levels observed in specific brain regions. The data were retrieved from the GTEx database on 3 October 2024.

**Figure 5 ijms-25-11043-f005:**
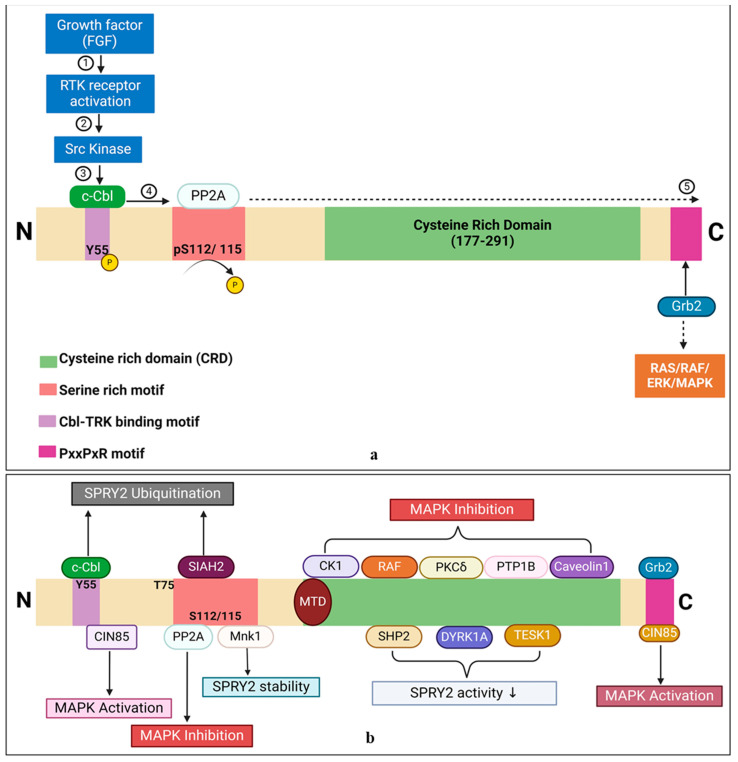
Structure and interacting sites of hSPRY2-binding proteins and their response in RTK signaling. (**a**) Structure and activation of hSPRY2 protein; GFs-induced activation of RTK activates Src kinase, and Src kinase phosphorylates the Y55 residue in the Cbl-TRK binding motif of SPRY2, followed by recruitment of PP2A, which dephosphorylates Ser112 and 115; dephosphorylation of these Ser residues resulted in conformational change at the C-terminal proline-rich motif and promoted the binding of growth factor receptor bound protein 2 (GRB2). The N-terminal SH3 domain of GRB2 is constitutively linked to the GTPase Son of Sevenless (SOS). The SH2 domain of GRB2 attaches itself to phosphorylated tyrosine residues on GF receptors, connecting receptor activation to the SOS-RAS-MAP kinase signaling cascade. The binding of GRB2 on the C-terminal of SPRY2 prevents its interaction with SOS, which is required for downstream ERK activation. (**b**) SPRY2-binding partners and their responses in RTK signaling. Casitas B lineage lymphoma (c-Cbl) and Siah E3 Ubiquitin Protein Ligase 2 (SIAH2) bind to the SPRY2-N-terminal and are responsible for SPRY2 ubiquitinylation; however, the binding of Phosphoprotein phosphatase 2A (PP2A) inhibits the ERK downstream signaling. The binding of CIN85 on the C or N-terminal inhibits SPRY2 negative regulation of the MAPK cascade and activates MAPK signaling. Phosphorylation of Ser112/115 stabilizes the SPRY2 protein; however, dephosphorylation of these residues promotes the binding of GRB2 and rapidly accelerated fibrosarcoma (RAF) that inhibit the downstream signaling of ERK. Other binding partners are also shown in (**b**) that bind to the C-type carbohydrate recognition domain (CRD) domain and inhibit the RTK or MAPK pathway. (The concept of (**a**) was adopted from (Guy et al., 2009) [24] and (**b**) from (Edwin et al., 2009) [25], modified and recreated by BioRender).

**Figure 6 ijms-25-11043-f006:**
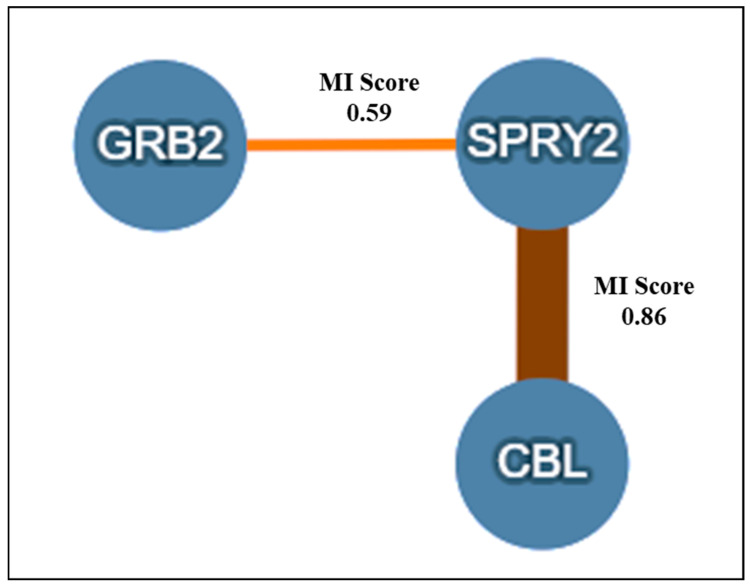
The direct protein–protein interaction analysis was graphically represented using the IntAct databases, incorporating experimental evidence. SPRY2 directly interacts with proteins such as GRB2 and Cbl, exhibiting MI scores (number of interactions) of 0.59 and 0.86, respectively.

**Figure 7 ijms-25-11043-f007:**
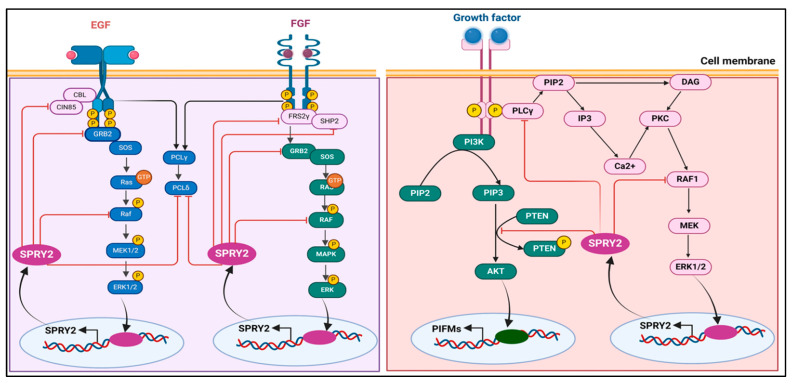
A basic schematic representation of feedback regulation of FGF/EFGF and PI3K/PIP2 signaling by SPRY2. After being induced, SPRY2 binds to the adopter protein, translocates to the membrane, and undergoes Tyr55 phosphorylation and shows the inhibitory activity on signaling cascades by targeting different signaling molecules, as shown in the figure. The concept of the figure was adopted from Mason et al., 2006, and Y.-W. Zhang et al., 2013 [48,49], and modified and recreated by BioRender.

**Figure 8 ijms-25-11043-f008:**
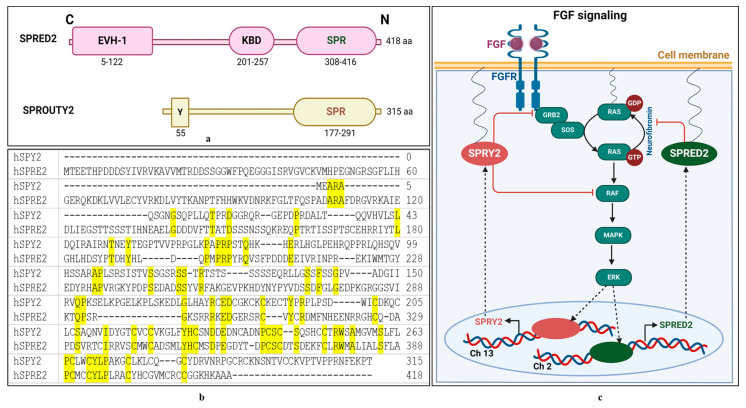
Characterization of human SPRED2 (hSPRE2) and SPROUTY 2 (hSPY2) protein. (**a**) Comparison of the domain structure of SPRED2 and SPROUTY2. (**b**) Alignment of the amino acid sequence of the SPRED2 and SPROUTY2 proteins (similar amino acid sequences are highlighted). (**c**) Mechanism of action of hSPRED2 and hSPROUTY2 on the ERK pathway. ((**a**) was created on PowerPoint, (**b**) is an alignment from UniProt, and (**c**) is adopted from Lorenzo and McCormick, 2020 [57], and is modified and recreated by BioRender).

**Table 1 ijms-25-11043-t001:** Summary of the human SPROUTY protein family. (Data retrieved from https://www.uniprot.org/).

Characteristics	SPROUTY 1	SPROUTY 2	SPROUTY 3	SPROUTY 4
UniProt ID	O43609	O43597	O43610	Q9C004
Gene	SPRY1	SPRY2	SPRY3	SPRY4
Chromosomal location	4q26.1	13q31.1	XqPAR2	5q31.3
Amino acid (aa) no and MW	319 aa 35,122 Da	315 aa 34,688 Da	288 aa 31,222 Da	299 aa 32,541 Da
Tissue specificity	Primarily in adipose tissue	Low tissue specificity	Brain	Adipose tissue and brain
Molecular function	Developmental protein	Developmental protein	Developmental protein (Primarily Neurogenesis)	Developmental protein
Interacting partner	TESK1, and CAV1; Forms heterodimers with SPRY2	GAB1, METTL13, SPRY2, RAF1, TESK1, GRB2, PPP2R1A/PP2A-A, PPP2CA/PP2A-C, c-Cbl, and CAV1; Forms heterodimers with SPRY1	TESK1, USP11, and CAV1	TESK1, RAF1, and CAV1
Conserved tyrosine residue	Y53	Y55	Y27	Y52
SPR domain for RAS regulation	183–295 aa	177–291 aa	154–260 aa	166–273 aa
Role in the RTK signaling pathway	Inhibits FGF-mediated phosphorylation of ERK1/2	Inhibits FGF-mediated phosphorylation of ERK1/2	Inhibits EGF-mediated activation of erk1/2	Inhibits Ras-independent activation of RAF1
Alphafold protein structure	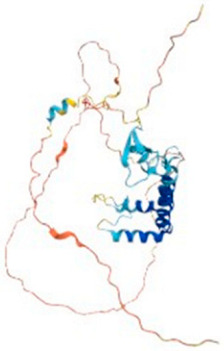	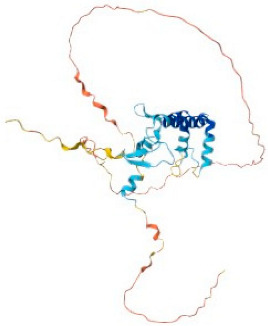	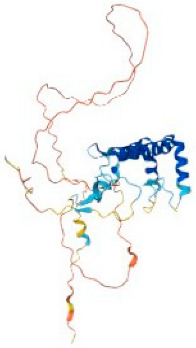	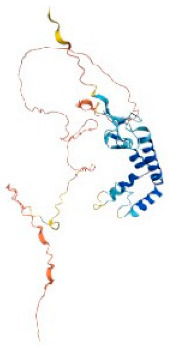

**Table 2 ijms-25-11043-t002:** SPROUTY 2 interacting proteins and their mechanism of action and effect on SPRY2 function.

Protein	Class/Family	Localization	Mechanism	Effect on SPRY2	References
c-Cbl	E3 ubiquitin–protein ligase	Plasma membrane	Through their SH2-like domain, c-Cbl engages with the Tyr55 of SPRY2	Leads ubiquitin-linked SPRY2 demolition	[26,27,28,29]
SIAH2	Ubiquitin Ligases	Cytoplasm	SIAH2 binds to the N-terminal of SPRY2 in a Tyr phosphorylation-independent manner	SIAH2 regulates the number of SPRY proteins by ubiquitinoylation	[30,31]
CIN85	Adaptor proteins	Plasma membrane	SH3 domains of CIN85 bind to Pro/Arg-rich motifs of the N and C-terminal of SPRY2	It binds to SPRY2 and controls the clustering of c-Cbl	[32]
Mnk1	Ser kinase	Cytoplasm	Phosphorylates the Ser112 and Ser115/121 of SPRY2	Modulates the tertiary structure of SPRY2 essential for SPRY2 activity	[33]
TESK1	Ser/Thr kinase	Cytoplasm	Intrudes with SPRY2/GRB2 binding and dephosphorylation of critical Ser residues by PP2A	Diminishes the SPRY2 inhibition effect on downstream signaling	[34]
DYRK1A	Ser/Thr and Tyr kinase	Cytoplasm	Interacts with SPRY2 and phosphorylate Thr75	Diminishes the SPRY2 inhibition of downstream signaling	[35]
PTEN	Phosphatase	Cytoplasm	SPRY2 expression positively regulated the PTEN activity by enhancing its amount and also reducing its phosphorylation	Inhibits AKT downstream signaling by increasing SPRY2 expression	[36]
PP2A	Ser/Thr protein phosphatase	Nucleus and cytoplasm	PP2A dephosphorylates Ser112 and Ser115	Dephosphorylation of Ser residues influences phosphorylation of Tyr55 leading to stimulation of SPRY2 activity	[26]
SHP2	Ser/Thr protein phosphatase	Nuclear/cytoplasm	Active SHP2 resulted in the dephosphorylation of Tyr55	Reduced SPRY2 binding to GRB2, which lessens SPRY2’s inhibitory effects on RTK signaling	[37]
PTP1B	Phosphatases	Cytoplasmic face of the endoplasmic reticulum	Reduced Tyr phosphorylation of p130Cas	PTP1B reduced SPRY2’s capacity to impede cell migration but not cell proliferation	[38]
TESK1	Phosphatases	Cytoplasm	TESK1 inhibits SPRY2 translocation to membrane ruffles and also inhibits PP2A-mediated dephosphorylation of Ser residues on SPRY2	Tesk1 inhibits SPRY2’s inhibitory mechanism by preventing it from interacting with the adaptor protein GRB2	[34]
Caveolin-1	Cholesterol-binding protein	Plasma membrane-associated protein	SPRY2 associated with Cav-1 by CRD	Inhibits EGF-induced p42/44 ERK inhibition	[39]
GRB2	Member of GRB2/sem-5/DRK family	Cytoplasm	SH3 domain of GRB2 binds to SPRY2	Prevents ERK activation upstream of RAS	[40]
PKC⸹	Member of PKC family	Cytoplasm	PhasphorylatedTyr55 facilitate SPRY2 binding to PKC⸹	Inhibites ERK phosphorylation and consequently, inhibit RAS/ERK pathway	[41]
Casein kinase-1	Ser/Thr-protein kinase	Cytoplasm	Controls SPRY2 in a phosphorylation-dependent manner	CK1 activity and recruitment are necessary for SPRY2 function	[16]
RAF	Ser/Thr-protein kinase	Cytoplasm	Phosphorylation of Ser111 and S120 residue of SPRY2 is required for Raf binding	RAF kinase activity was significantly decreased upon attachment to SPRY2; inhibit RAS/MAP kinase pathway	[42,43]

**Table 3 ijms-25-11043-t003:** Role of SPROUTY 2 in various cell signaling cascades.

Signaling Cascade	Receptor	Function	SPRY2 Activation	Mechanism of Action	Cascade Downstream Function	References
FGF signaling	RTK	Antagonist for FGF signaling	Phosphorylation of Tyr-227	The conserved Tyr residue at the N-terminal of SPRY2 binds to GRB2, and inhibits FGF downstream signaling cascade	Endothelial growth, migration, and morphogenesis	[16]
EGF signaling	RTK	Antagonist for EGF signaling	Phosphorylation of Tyr-55	SPRY2 binds to adaptor CIN85 and leads to EGFR degradation or endocytosis	Endothelium development	[50,51]
VEGF signaling	RTK	Antagonist for VEGF signaling	Tyr phosphorylation of SPRY is not required	SPRY binds to RAF1 and blocks further VEGF signaling	Vascular branching in lung development	[52]
BDNF signaling	RTK	Antagonist for BDNF signaling	Phosphorylation of SPRY on critical Tyr residues	Inhibits neurite formation and the expression of neurofilament light chain; overexpression of SPRY2 induces neuronal cell death	Neuronal differentiation and neurite outgrowth	[53]
IFN signaling (Jak-Stat pathways)	IFN receptor	Antagonist of p38 MAPK pathway	Phosphorylation of SPRY2 on Ser-112 and Ser-121	Suppresses the p38 MAPK	Inhibits cell proliferation and suppresses tumor formation; antiviral and antineoplastic properties	[22]
PDGFsignaling	PDGF receptor	An antagonist of RAF-MEK-ERK signaling	Phosphorylation of Tyr by c-Src	PDGF activated c-Src, phosphorylated the SPRY2 at Tyr55, and inhibited PDGF-induced ERK activation	NIH3T3 cell proliferation (in vitro study)	[54]
GDNF signaling (Glial cell-derived neurotrophic factor)	RET receptorTyr kinase	Antagonist of ERK pathway	Phosphorylation of Tyr 55	Inhibits GDNF-induced ERK activation	Regulation of neural cell proliferation, differentiation, and migration of enteric neural crest cells	[55]
AKT pathway		Antagonist of AKT pathway	SPRY2 decreases PTEN phosphorylation on Ser380, Thr382 and Thr383	Unphosphorylated PTEN is more stable and blocks PI3K/AKT signaling	Suppressed the cell proliferation and production of proinflammatory cytokines and matrix metalloproteinases	[56]

**Table 4 ijms-25-11043-t004:** Role of SPRY2 in neurodevelopmental disorders.

Aim	SPROUTY Mutation/Downregulation/Overexpression	Study on	Techniques	Description	Effect on the Signaling Pathway	References
Study the role of the SPRY2 C-terminal SH3-binding motif	Y55F and R309A	Pheochromocytoma cells (PC-12 cells)	Immunoblot and Neurite growth	Double-point mutation inhibits SPRY2 Binding activity with both GRB2 and c-Cbl	MAPK/ERK not inhibited	[73]
	SPRY2 P314A mutant	PC-12 cells	Immunoblot and Neurite growth	Mutant SPRY2 showed less binding to GRB2	inhibition of MAPK/ERK pathway	
Study the siRNA-based inhibition of SPRY2	Deletion of SPRY2 by siRNA technology	PC-12, C6 glioblastoma cells and NIH3T3 fibroblasts cells	qRT-PCR, Western blot, and neurite outgrowth	Down-regulation of SPRY 1, -2, and -4 increased the neurite length of PC12 cells	Activation of RAS/ERK pathway in response to SPRY2 downregulation	[45]
Role of SPRY2 and SPRY4 on embryonic morphogenesis and regulation	KO of SPRY2 and SPRY4 (Double KO by siRNAs)	Mice	KO mice lung bud and brain culture and mice phenotype	SPRY2 KO mice showed severe defects in craniofacial, limb, and lung morphogenesis	SPRY2/SPRY4 important for modulating FGF8/FGF10 signaling	[74]
Role of SPRY2 and -4 on the regulation of axon outgrowth	KO of SPRY2 and SPRY4 (Double KO by siRNAs)	Hippocampal neurons of BALB/c mice	Double KO by siRNAs; qRT-PCR, Axonal growth, and neuronal morphologies assay	Downregulation of SPRY2 and -4 promote axon growth	FGF signaling is regulated by both SPRY2 and 4	[75]
Role of SPRY2 in neurogenesis and stress responsiveness	Downregulation of SPRY2 expression	Male Sprague Dawley rats	Behavior activity	Trigger neurogenesis and improve stress resilience indicators, such as the accelerated eradication of conditioned fear	Decrease SPRY2 expression; increase FGF2 signaling	[65]
Effect of SPRY2 in growth and differentiation of human neuroblastoma cells through GDNF-ERK signaling	Transfection of SPRY2 (and others) in HEK-293 cells	SPRY2-deficient mice	HEK 293T cells; TGW cell proliferation assay	During development, SPRY2 adversely controls the proliferation of enteric neural crest cells	Expression of SPRY2 inhibits GDNF-induced ERK activation	[55]
Role of SPRY2 in BDNF-induced neuronal differentiation and survival	Overexpression and KO of SPRY2	Mice neuron	Neuritogenesis assay, immunoblot	Overexpression of SPRY2 induces neuronal cell death, whereas downregulation of SPRY2 promotes neuronal survival	SPRY2 downregulates BDNF-driven signaling	[53]
Role of SPRY2 in eye development	Overexpression and KO of SPRY2	Xenopus embryos	Histology and immunohistochemistry	KO of SPRY2 inhibit retinal progenitor’s growth resulted in small eye size	SPROUTY2 suppresses MAPK activity	[76]
Role of SPRY2 in eye and lens development	Overexpression	Mice	Eye histology and lens differentiation study	Overexpression of SPRY2 in the lens resulted in reduced lens and eye size	SPRY2 protein interferes in the FGF-and EGF-mediated MAPK/ERK1/2 signaling	[77]
Role of SPRY2 in axon outgrowth and regeneration	Downregulation	SPRY2^−/+^ and SPRY2^−/−^ Mice	Neuron culture, immunocytochemistry and histochemistry	SPRY2 limited the axon outgrowth and nerve regeneration	Downregulate MAPK/ERK signaling	[69]
Role of SPRY2 in axon regerneration	Downregulation	Spry2^−/−^ KO mice	Immunocytochemistry and axon growth assay	Downregulation of SPRY2 promotes axon regeneration	Suppression of SPRY2 expression increased the ERK activation	[78]

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
