# Peer review of "SPROUTY2, a Negative Feedback Regulator of Receptor Tyrosine Kinase Signaling, Associated with Neurodevelopmental Disorders: Current Knowledge and Future Perspectives"

_ijms, 2024, doi:10.3390/ijms252011043_

Round 1
Reviewer 1 Report
Comments and Suggestions for Authors
The authors provided the literature overview about structural and functional features of SPRY2 (SPROUTY2) protein, its main interacting partners crucial in receptor tyrosine kinase (RTK) signaling pathways, and how are they involved, both, in normal and disease states. The Review is of sufficient significance; however, several issues need to be addressed.
Concerns:
1. Although Title indicates that the focus of the review will, also, be on therapeutic implications, the text of the manuscript provides limited information about this subject, which indicates that the title does not correspond to the rest of the study. The whole section (subsection) about therapeutic implications should either be fully developed or the title should be revised to align with the content of the manuscript. Also, the last sentence of the section Abstract should be modified to ensure consistency with the rest of the manuscript.
2. Article by Hausott and Klimaschewski, 2019, states that Spry1, Spry2, and Spry4 are detected in the neopallial cortex, cranial flexure, and cerebellum, while in postnatal mouse brains, Spry2 and Spry4 are the major CNS isoforms in neuronal and glial cells of the cerebellum, cortex, and hippocampus. The authors, also, state that Spry3 is only detected at low levels in the brain. On the other hand, in the Table 1., Nidhi Puranik, Jeong Ho Jeong and Minseok Song, stated that Spry3 and 4 are expressed in the brain. Provide the explanation for the mismatch of the information.
Hausott B, Klimaschewski L. Sprouty2-a Novel Therapeutic Target in the Nervous System? Mol Neurobiol. 2019, 56(6):3897-3903. doi: 10.1007/s12035-018-1338-8. PMID: 30225774; PMCID: PMC6505497.
3. The citation of Table 3. and 4., as well as Figure 4. are lacking in the text. The Tables and Figure should be citied in the paragraphs that are focused on the topics presented in the Tables and Figure. Moreover, the Figure 1b is not provided, most likely the authors mismatched Figure 1b with Figure 3b. The manuscript should be read carefully and this issue should be resolved.
4. For all listed abbreviations in the manuscript the comprehensive explanations should be provided, including PI-PLC, etc.
5. Although the legends are detailed, the encompassing explanations of the listed abbreviations depicted in the Figures should be provided along with the abbreviations.
6. When citing authors and studies in the text ‒ somewhere in the text the first author and et al., is citied followed by the year, while in other places the year is lacking. It is necessary to standardize the way references are cited in the text.
7. The section Conclusion and future perspectives is not thorough enough, it should be more precise and provide stronger conclusions and offer some directions for future studies.
Author Response
Comments and Suggestions for Authors
The authors provided the literature overview about structural and functional features of SPRY2 (SPROUTY2) protein, its main interacting partners crucial in receptor tyrosine kinase (RTK) signaling pathways, and how are they involved, both, in normal and disease states. The Review is of sufficient significance; however, several issues need to be addressed.
Response: Thank you so much for providing your valuable time in reviewing the manuscript. We tried to incorporate all your comments in the revised version of the manuscript.
Concerns:
- Although Titleindicates that the focus of the review will, also, be on therapeutic implications, the text of the manuscript provides limited information about this subject, which indicates that the title does not correspond to the rest of the study. The whole section (subsection) about therapeutic implications should either be fully developed or the title should be revised to align with the content of the manuscript. Also, the last sentence of the section Abstractshould be modified to ensure consistency with the rest of the manuscript.
Response: Thank you for your insightful suggestion. We agree with that. The title and abstract are modified accordingly.
Title: SPROUTY2, A Negative Feedback Regulator of Receptor Tyro-sine Kinase Signaling, Associated with Neurodevelopmental Disorders: Current Knowledge and future prospective.
Abstract: deleted- Meanwhile, SPRED2, a SPRY-like protein that has a conserved C-terminal SPR domain for RAS regulation, is well established in BDNF signaling and various neurodevelopmental disorders, in-cluding anxiety, stress-associated pathologies, schizophrenia, and obsessive-compulsive disorder
Added-Recent research using a variety of small animal models suggests that SPRY2 has substantial therapeutic promise for treating a range of neurological conditions. This is explained by its function as an intracellular ERK signaling pathway inhibitor, which is connected to a variety of neuronal activities. By modifying this route, SPRY2 may open the door to novel therapeutic approaches for these difficult-to-treat illnesses.
- Article by Hausott and Klimaschewski, 2019,states that Spry1, Spry2, and Spry4 are detected in the neopallial cortex, cranial flexure, and cerebellum, while in postnatal mouse brains, Spry2 and Spry4 are the major CNS isoforms in neuronal and glial cells of the cerebellum, cortex, and hippocampus. The authors, also, state that Spry3 is only detected at low levels in the brain. On the other hand, in the Table 1., Nidhi Puranik, Jeong Ho Jeong and Minseok Song, stated that Spry3 and 4 are expressed in the brain. Provide the explanation for the mismatch of the information. (Hausott B, Klimaschewski L. Sprouty2-a Novel Therapeutic Target in the Nervous System? Mol Neurobiol. 2019, 56(6):3897-3903. doi: 10.1007/s12035-018-1338-8. PMID: 30225774; PMCID: PMC6505497).
Response: Thank you for your valuable input. Yes, according to Hausott and Klimaschewski, 2019, Spry3 is only detected at low levels in the brain of mouse. However, the data presented in Table 1 is exported from UniProt for human SPRY proteins where it is mentioned that (Tissue specificity) SPRY3 is Widely expressed; particularly in the fetal tissues. Expressed in the brain with expression the highest in Purkinje cells in the cerebellum. (https://www.uniprot.org/uniprotkb/O43610/entry)
- The citation of Table 3. and 4., as well as Figure 4.are lacking in the text. The Tables and Figure should be citied in the paragraphs that are focused on the topics presented in the Tables and Figure. Moreover, the Figure 1bis not provided, most likely the authors mismatched Figure 1b with Figure 3b. The manuscript should be read carefully and this issue should be resolved.
Response: Agree with reviewer point. All the figures and tables are properly cited in revised MS.
Table 3- The aforementioned studies demonstrate that SPRY2 not only regulates the RTK sig-naling pathway but also modulates additional critical pathways involved in cell growth and development. A comprehensive overview of the role of SPRY2 across various signal-ing pathways, along with its mechanisms of action, is summarized in Table 3.
Table 4- Overall, a variety of in vitro and in vivo studies demonstrate the potential of SPRY2 in regulating normal neuronal cell development. Alterations in the expression or mutations of SPRY2 proteins can disrupt signaling cascades, leading to neurodevelopmental disor-ders. A comprehensive summary of the key in vitro and in vivo studies is presented in Table 4.
Figure 4-After RTK activation, they move from the cytosol to the cell membrane, where they are phosphorylated at Tyr55 residues and attach attached via palmitoylation as shown in Figure 4. A simplified schematic illustrates the feedback regulation of FGF/EFGF and PI3K/PIP2 signaling by SPRY2.
For mismatch- corrected in the text as well in figure.
- For all listed abbreviations in the manuscript the comprehensive explanations should be provided, including PI-PLC, etc.
Response: Agree with reviewer point. Following added in revised MS.
PI-PLC, Phosphatidylinositol-specific phospholipase C
GRB2, growth factor receptor bound protein 2
c-Cbl, Casitas B lineage lymphoma
PP2A, Phosphoprotein phosphatase 2A
SIAH2, Siah E3 Ubiquitin Protein Ligase 2
PKC⸹, Protein kinase C delta
TESK1, Testis Associated Actin Remodelling Kinase 1
PTEN, Phosphatase and tensin homolog
DYRK1A, dual specificity tyrosine phosphorylation regulated kinase 1A
MNK1, MAP kinase-interacting serine/threonine-protein kinase 1
CIN85, Cbl-interacting 85-kDa protein
FRS2α, fibroblast growth factor receptor 2
- Although the legends are detailed, the encompassing explanations of the listed abbreviations depicted in the Figures should be provided along with the abbreviations.
Response: Thank you for your feedback. Agree with the reviewer's point. As figures 1 and 3 have the most abbreviated form, their legends have been modified accordingly.
Figure 1. Fibroblast growth factors (FGFs) interact with the FGF receptor domain and facilitate receptor dimerization autophosphorylation, which in turn attracts other secondary signaling molecules such as Phospholipase C Gamma (PLCγ), fibroblast growth factor receptor 2 (FRS2α), and Growth factor receptor-bound protein 2 (GRB2) and assembles them in signaling complexes. A representation of the three primary downstream signaling complexes–– the Ras/MEK/MAPK/ERK, PI3K/AKT, and PLCγ pathways–– is shown. MAPK/ERK is translocated in the nucleus and activates the gene expressions responsible for cell proliferation and differentiation as well as for feedback inhibition.
Figure 3. Structure and interacting sites of hSPRY2 binding proteins and their response in RTK signaling. a) Structure and activation of hSPRY2 protein; GFs-induced activation of RTK activates Src kinase, and Src kinase phosphorylates the Y55 residue in Cbl-TRK binding motif of SPRY2 followed by recruitment of PP2A which dephosphorylates Ser112 and 115; dephosphorylation of these Ser residues resulted in conformational change at the C-terminal proline-rich motif and promoted the binding of growth factor receptor bound protein 2 (GRB2). The N-terminal SH3 domain of GRB2 is constitutively linked to the GTPase Son of Sevenless (SOS). The SH2 domain of GRB2 attaches itself to phosphorylated tyrosine residues on GF receptors, connecting receptor ac-tivation to the SOS-RAS-MAP kinase signaling cascade. The binding of GRB2 on the C-terminal of SPRY2 prevents its interaction with SOS, which is required for downstream ERK activation. b) SPRY2 binding partners and their responses in RTK signaling. Casitas B lineage lymphoma (C-Cbl) and Siah E3 Ubiquitin Protein Ligase 2 (SIAH2) bind to the SPRY2-N-terminal and are re-sponsible for SPRY2 ubiquitinylation; however, the binding of Phosphoprotein phosphatase 2A (PP2A) inhibits the ERK downstream signaling. The binding of CIN85 on the C or N-terminal in-hibits the SPRY2 negative regulation of the MAPK cascade and activates the MAPK signaling. Phosphorylation of Ser112/115 stabilizes the SPRY2 protein; however, dephosphorylation of these residues promotes the binding of GRB2 and rapidly accelerated fibrosarcoma (RAF) that inhibit the downstream signaling of ERK. Other binding partners are also shown in Figure 1b 3b that bind to the C-type carbohydrate recognition domain (CRD) domain and inhibit the RTK or MAPK pathway.
- When citing authors and studies in the text ‒ somewhere in the text the first author and et al., is citied followed by the year, while in other places the year is lacking. It is necessary to standardize the way references are cited in the text.
Response: Agree with reviewer point. Modification is done accordingly.
DaSilva et al. (2006) [20]
Hanafusa et al. (2002) [8]
- The section Conclusion and future perspectivesis not thorough enough, it should be more precise and provide stronger conclusions and offer some directions for future studies.
Response: Thank you for your valuable input. The conclusion and future perspectives has been modified.
Deleted- In the brain, ERK is involved in learning and memory because it facilitates the release of synaptic transmitters and induces the tran-scription of genes relevant to plasticity. SPROUTYs have considerable potential in the treatment of various neurological illnesses because they are intracellular inhibitors of the ERK pathway.
Studies on SPRY proteins, their interactions, post-translational modifications, and cellular levels are crucial for understanding their biological actions. Understanding the corollary of important connections is essential for understanding SPRY isoforms. The ab-sence of enzymatic activity in the protein makes it challenging to create chemical com-pounds that can precisely and directly modulate SPRY function as a therapeutic target. However, a practical method for assessing the therapeutic potential of indirectly boosting the ERK is to interfere with SPRY expression using gene therapy or siRNA treatment.
Added- Sprouty2 is expressed in brain tissue and serves as a critical regulatory protein of the RTK signaling pathway. Studies have demonstrated that SPRY2 is upregu-lated in various experimental models of neuronal degeneration and regeneration. Its in-creased expression under pathological conditions positions SPRY2 as a promising phar-macological target for enhancing intracellular signaling activities, particularly within the ERK pathway, in affected neurons and activated astrocytes. Since these preliminary dis-coveries, the number of biological processes and pathways regulated by SPRY proteins has steadily increased.
Thank you for reviewing our article. We have addressed all the changes and corrections suggested by the reviewer. Please let us know if any further modifications or updates are needed.

Reviewer 2 Report
Comments and Suggestions for Authors
This is an extensive and interesting review that describes how the modulation of tyrosine kinase receptors by SPRY2 regulates cell proliferation, growth, and differentiation. The authors effectively summarize the function, interaction, and role of the SPROUTY family members in RTK signaling pathways, using tables to enhance understanding. Additionally, the molecular mechanisms of SPRY2 are described in detail.
However, a few comments should be considered:
1. The summary should be clearer. For example, what does the inhibition of RAS and the reduction of RAS/ERK mean (line 13)?
2. Figure 1 needs to be explained in detail.
3. On line 161, it should read 3b instead of 1b.
4. The expression of this modulator throughout the lifespan and its role in some neuropathologies should be indicated.
5. It is important to specify in which types of neuronal cells the pathways modulated by SPRY2 are expressed.
6. This reference should include: Felfly H, Klein OD. Sprouty genes regulate proliferation and survival of human embryonic stem cells. Sci Rep. 2013;3:2277. doi: 10.1038/srep02277. PMID: 23880645; PMCID: PMC3721083.
Author Response
Comments and Suggestions for Authors
This is an extensive and interesting review that describes how the modulation of tyrosine kinase receptors by SPRY2 regulates cell proliferation, growth, and differentiation. The authors effectively summarize the function, interaction, and role of the SPROUTY family members in RTK signaling pathways, using tables to enhance understanding. Additionally, the molecular mechanisms of SPRY2 are described in detail.
Response: Thank you so much for providing your valuable time in reviewing the manuscript. We tried to incorporate all your comments in the revised version of the manuscript.
However, a few comments should be considered:
- The summary should be clearer. For example, what does the inhibition of RAS and the reduction of RAS/ERK mean (line 13)?
Response: Thank you for your insightful suggestion. We agree with that.
SPROUTY2 (SPRY2) is a potent negative regulator of receptor tyrosine kinase (RTK) signaling that binds to GRB2 during RTK activation and inhibits the GRB2-SOS complex, which inhibits RAS activation and attenuates the downstream further reduces RAS/ERK signaling cascade.
- Figure 1 needs to be explained in detail.
Response: Thank you for your valuable input. Figure 1 is explained in the revised MS.
FGFs interact with their specific receptors, facilitating receptor dimerization and autophosphorylation. This activation process recruits secondary signaling molecules, including PLCγ, FRS2α, and GRB2, which contributes to the formation of signaling complexes. The interaction of FGFs with their receptors activates three primary downstream signaling pathways: the Ras/MEK/MAPK/ERK pathway, the PI3K/AKT pathway, and the PLCγ pathway. The MAPK/ERK pathway subsequently translocates to the nucleus, where it regulates gene expression related to cell proliferation, differentiation, and feedback inhibition.
- On line 161, it should read 3b instead of 1b.
Response: Agree with the reviewer. The correction has been done in both texts as well in the figure.
- The expression of this modulator throughout the lifespan and its role in some neuropathologies should be indicated.
Response: Thank you for your valuable feedback. I appreciate your insights. However, it's important to note that there hasn’t yet been a comprehensive study detailing the specific expression profile of SPRY2 across the lifespan. Exploring its modulation throughout different developmental stages remains essential. Additionally, all the SPRY2-associated neuropathologies are discussed in Section 1.4 and summarized in Table 4.
- It is important to specify in which types of neuronal cells the pathways modulated by SPRY2 are expressed.
Response: Thank you for your feedback. The following part is added in section 1.2.
SPRY2 protein is expressed in the brain and various neuronal cell types, including neurons, astrocytes, oligodendrocytes, and neural stem cells. In these cells, SPRY2 plays a key role in modulating signaling pathways that are critical for neuronal development, survival, and function.
- This reference should include: Felfly H, Klein OD. Sprouty genes regulate proliferation and survival of human embryonic stem cells. Sci Rep. 2013;3:2277. doi: 10.1038/srep02277. PMID: 23880645; PMCID: PMC3721083.
Response: The references cited in the revised MS.
SPRY2 also plays a crucial role in human embryonic stem cell (hESC) self-renewal, with low levels reducing survival and proliferation. However, SPRY2 knockdown cells remain responsive to growth factors, showing increased cell numbers in response to higher concentrations of FGF2 and EGF (Felfly & Klein, 2013).
Thank you for reviewing our article. We have addressed all the changes and corrections suggested by the reviewer. Please let us know if any further modifications or updates are needed.

Reviewer 3 Report
Comments and Suggestions for Authors
Summary
The authors have provided a comprehensive and detailed review of SPROUTY genes and proteins. However, I recommend enhancing the graphical content of the manuscript by incorporating additional bioinformatic analyses. Additionally, I highly recommend changing the statement of the manuscript from "Article" to "Review". I recommend a 'Major Revision' to allow you sufficient time to address the suggested improvements.
Introduction
Lines 31-50: This section requires additional references to support the various genes and pathways mentioned. It is important to cite relevant studies for each pathway to strengthen the scientific foundation of this review.
Figures 1, 2, 3, 4, and 5: The descriptions provided for the figures are well detailed and informative. Nevertheless, you must specify the source of the images. For example, if they are adapted or modified from a particular study or a molecular pathway database, this should be clearly mentioned in the figure legends.
Table 1: The description of Table 1 requires further elaboration to improve clarity and provide more context.
Regarding the percentage of identity among the various SPROUTY proteins, it would be valuable to calculate this using tool such as the UniProt database or BioEdit. To enhance the visual and scientific quality of your review, I recommend including the following:
- A heatmap illustrating the percentage identity between different SPROUTY proteins.
- A sequence alignment (similar to Figure 5) to identify regions of divergence among the proteins.
- A graphical representation of the structural models of these proteins, which can be generated using AlphaFold models or protein modeling software like UCSF Chimera X or PyMol.
You can find the methodology for these bioinformatic analyses in this manuscript (https://doi.org/10.3390/cimb46070383). If You deemed helpful, should be included in Your manuscript.
Table 2: How were the SPROUTY 2 interacting proteins identified? Were these derived solely from the references provided, or was a specific database used? I recommend utilizing STRING, BioGrid, or IntAct databases to strengthen your findings and including a visual representation of the interaction network (exported as a PNG) in the manuscript.
You have discussed gene expression on multiple occasions, but it would be helpful to specify which organs or tissues these genes are expressed in. Databases such as HPA (Human Protein Atlas) or GTEx can provide valuable insights into tissue-specific expression patterns, which should be incorporated into the discussion.
Author Response
Comments and Suggestions for Authors
Summary
The authors have provided a comprehensive and detailed review of SPROUTY genes and proteins. However, I recommend enhancing the graphical content of the manuscript by incorporating additional bioinformatic analyses. Additionally, I highly recommend changing the statement of the manuscript from "Article" to "Review". I recommend a 'Major Revision' to allow you sufficient time to address the suggested improvements.
Response: Thank you so much for providing your valuable time in reviewing the manuscript. We tried to incorporate all your comments in the revised version of the manuscript.
Introduction
Lines 31-50: This section requires additional references to support the various genes and pathways mentioned. It is important to cite relevant studies for each pathway to strengthen the scientific foundation of this review.
Response: Thank you for your insightful suggestion. We agree with that. The following citations are added at the appropriate place.
Batool, Z., Azfal, A., Liaquat, L., Sadir, S., Nisar, R., Inamullah, A., Faiz Ghalib, A. U., & Haider, S. (2023). Receptor tyrosine kinases (RTKs): from biology to pathophysiology. Receptor Tyrosine Kinases in Neurodegenerative and Psychiatric Disorders, 117–185. https://doi.org/10.1016/B978-0-443-18677-6.00012-9
Castrén, E., & Monteggia, L. M. (2021). Brain-Derived Neurotrophic Factor Signaling in Depression and Antidepressant Action. Biological Psychiatry, 90(2), 128–136. https://doi.org/10.1016/J.BIOPSYCH.2021.05.008
Diez del Corral, R., & Morales, A. V. (2017). The Multiple Roles of FGF Signaling in the Developing Spinal Cord. Frontiers in Cell and Developmental Biology, 5(JUN). https://doi.org/10.3389/FCELL.2017.00058
Kiyatkin, A., Rosenburgh, I. K. van A. van, Klein, D. E., & Lemmon, M. A. (2020). Kinetics of receptor tyrosine kinase activation define ERK signaling dynamics. Science Signaling, 13(645), 5267. https://doi.org/10.1126/SCISIGNAL.AAZ5267
Maruyama, I. N. (2014). Mechanisms of Activation of Receptor Tyrosine Kinases: Monomers or Dimers. Cells, 3(2), 304. https://doi.org/10.3390/CELLS3020304
Neben, C. L., Lo, M., Jura, N., & Klein, O. D. (2019). Feedback regulation of RTK signaling in development. Developmental Biology, 447(1), 71–89. https://doi.org/10.1016/J.YDBIO.2017.10.017
Pulivarthi, C. B., Choubey, S. S., Pandey, S. K., Gautam, A. S., & Singh, R. K. (2023). Receptor tyrosine kinases: an overview. Receptor Tyrosine Kinases in Neurodegenerative and Psychiatric Disorders, 45–77. https://doi.org/10.1016/B978-0-443-18677-6.00011-7
Wee, P., & Wang, Z. (2017). Epidermal Growth Factor Receptor Cell Proliferation Signaling Pathways. Cancers 2017, Vol. 9, Page 52, 9(5), 52. https://doi.org/10.3390/CANCERS9050052
Wu, P. K., Becker, A., & Park, J. I. (2020). Growth Inhibitory Signaling of the Raf/MEK/ERK Pathway. International Journal of Molecular Sciences 2020, Vol. 21, Page 5436, 21(15), 5436. https://doi.org/10.3390/IJMS21155436
Xie, Y., Su, N., Yang, J., Tan, Q., Huang, S., Jin, M., Ni, Z., Zhang, B., Zhang, D., Luo, F., Chen, H., Sun, X., Feng, J. Q., Qi, H., & Chen, L. (2020). FGF/FGFR signaling in health and disease. Signal Transduction and Targeted Therapy 2020 5:1, 5(1), 1–38. https://doi.org/10.1038/s41392-020-00222-7
Zhang, N., & Li, Y. (2023). Receptor tyrosine kinases: biological functions and anticancer targeted therapy. MedComm, 4(6), e446. https://doi.org/10.1002/MCO2.446
Figures 1, 2, 3, 4, and 5: The descriptions provided for the figures are well detailed and informative. Nevertheless, you must specify the source of the images. For example, if they are adapted or modified from a particular study or a molecular pathway database, this should be clearly mentioned in the figure legends.
Response: Thank you for your insightful suggestion. We agree with that. We have added appropriate information in the revised MS.
Figure 1: The figure concept was adopted from (Diez del Corral & Morales, 2017) ), modified and recreated by BioRender.
Figure 2 was created on power point presentation)
Figure 3: The concept of Figure 3a was adopted from (Guy et al., 2009) and 3b from (Edwin et al., 2009), modified and recreated by BioRender.
Figure 4: The concept of the figure was adopted from (Mason et al., 2006; Zhang et al., 2013) and modified and recreated by BioRender.
Figure 5: Figure 5a was created on PowerPoint, figure 5b is an alignment from UniProt, and figure 5c is adopted from (Lorenzo & McCormick, 2020) and modified and recreated by BioRender.
Table 1: The description of Table 1 requires further elaboration to improve clarity and provide more context.
Response: Thank you for your insightful suggestion. We appreciate the suggestion for elaboration. While the paper focuses primarily on Sprouty 2, Table 1 was included to provide fundamental points for comparison with other Sprouty proteins. We will clarify in the description that Table 1 serves as a brief overview
Regarding the percentage of identity among the various SPROUTY proteins, it would be valuable to calculate this using tool such as the UniProt database or BioEdit. To enhance the visual and scientific quality of your review, I recommend including the following:
- A heatmap illustrating the percentage identity between different SPROUTY proteins.
- A sequence alignment (similar to Figure 5) to identify regions of divergence among the proteins.
- A graphical representation of the structural models of these proteins, which can be generated using AlphaFold models or protein modeling software like UCSF Chimera X or PyMol.
Thank you for your insightful suggestion. The referenced paper has provided valuable insights into the utilization of various tools for protein-related analyses, and we have included this citation in our revised manuscript. This suggested article is not only relevant to the current manuscript but will also be valuable for our future research endeavors. Thank you once again for highlighting and recommending this resource.
Although we have successfully incorporated the sequence alignment and calculated the percentage identity among the Sprouty proteins, we have faced difficulties in generating the corresponding heatmap visualization that’s why the heatmap is not included in the revised MS.
Treccarichi, S.; Calì, F.; Vinci, M.; Ragalmuto, A.; Musumeci, A.; Federico, C.; Costanza, C.; Bottitta, M.; Greco, D.; Saccone, S.; et al. Implications of a De Novo Variant in the SOX12 Gene in a Patient with Generalized Epilepsy, Intellectual Disability, and Childhood Emotional Behavioral Disorders. Curr. Issues Mol. Biol. 2024, 46, 6407-6422. https://doi.org/10.3390/cimb46070383
We are currently exploring the software to effectively create the heatmap and will ensure that this analysis will be helpful in further papers. The following has been incorporated in the revised MS.
To further explore the SPRY family we have used various online available tools as mentioned in (Treccarichi et al., 2024). The UniProt database (https://www.uniprot.org/) served as the primary resource for retrieving detailed information on the functional regions and domain proteins. The UniProt database (https://www.uniprot.org/) was used to retrieve the SPRY proteins-related information (accessed on 2024/10/03) and sequence. In addition, the UniProt alignment tool was utilized to determine the percentage identity of SPRY2 compared to each SPRY protein 1, 3, and 4 and phylogenetic distance. This approach facilitated a comprehensive analysis of sequence similarity and functional conservation among the SPRY family shown in Figure 3. The SPRY protein family shows around 40%-55% similarity index.
Figure 3: Sequence similarity between Sprouty proteins. a) Comparison of the Alignment of the amino acid sequence of the Sprouty 1, 2,3, and 4 proteins (similar amino acid sequences are highlighted); b) percentage identity and c) phylogenetic tree.
You can find the methodology for these bioinformatic analyses in this manuscript (https://doi.org/10.3390/cimb46070383). If You deemed helpful, should be included in Your manuscript.
Table 2: How were the SPROUTY 2 interacting proteins identified? Were these derived solely from the references provided, or was a specific database used? I recommend utilizing STRING, BioGrid, or IntAct databases to strengthen your findings and including a visual representation of the interaction network (exported as a PNG) in the manuscript.
Response: Thank you for your insightful suggestion. For the analysis of interacting partners, we utilized the STRING database (the resulting image is not shown in the article); however, in the article, we focused exclusively on proteins for which published data demonstrate a role in neurodevelopment. This approach ensures that our discussion is grounded in established research.
As per your valuable suggestion, we have used IntAct databases for the protein-protein interaction score.
Figure 6: The direct protein-protein interaction analysis was graphically represented using the In-tAct databases, incorporating experimental evidence. SPRY2 directly interacts with proteins such as GRB2 and Cbl, exhibiting MI Scores (number of interactions) of 0.59 and 0.86, respectively.
(Not included in the article)
You have discussed gene expression on multiple occasions, but it would be helpful to specify which organs or tissues these genes are expressed in. Databases such as HPA (Human Protein Atlas) or GTEx can provide valuable insights into tissue-specific expression patterns, which should be incorporated into the discussion.
Response: Thank you for your insightful suggestion. As per your suggestions, we have analyzed the expression profile of SPRY2 in various tissues and the data with the figure has been incorporated in the revised MS.
SPRY2 protein exhibits low tissue specificity, being expressed in nearly all major tissues. Analysis of SPRY2 expression across various tissues was conducted using the Geno-type-Tissue Expression (GTEx) database (https://www.gtexportal.org/). The results indicate that while SPRY2 is present in almost all tissues, it is expressed at notably higher levels in the cerebellum and cerebellar hemisphere of the brain.
Figure 4: Bulk tissue expression of SPRY2 protein across various tissues. The data in-dicate the presence of SPRY2 in all examined tissues, with notably higher expression levels observed in specific brain regions. The data was retrieved from the GTEx data-base on 2024/10/03.
Thank you for reviewing our article. We have addressed all the changes and corrections suggested by the reviewer. Please let us know if any further modifications or updates are needed.

Round 2
Reviewer 1 Report
Comments and Suggestions for Authors
The majority of the raised concerns is addressed and the content of the manuscript is upgraded to meet the necessary standards for publications.
Reviewer 3 Report
Comments and Suggestions for Authors
Authors provided a comprehensive improved version of the manuscript.
Comments on the Quality of English LanguageEnglish is fine. I should replace "prospective" with "perspective" in the Title.